# Using Regional Sero-Epidemiology SARS-CoV-2 Anti-S Antibodies in the Dominican Republic to Inform Targeted Public Health Response

**DOI:** 10.3390/tropicalmed8110493

**Published:** 2023-11-04

**Authors:** Beatris Mario Martin, Angela Cadavid Restrepo, Helen J. Mayfield, Cecilia Then Paulino, Micheal De St Aubin, William Duke, Petr Jarolim, Emily Zielinski Gutiérrez, Ronald Skewes Ramm, Devan Dumas, Salome Garnier, Marie Caroline Etienne, Farah Peña, Gabriela Abdalla, Beatriz Lopez, Lucia de la Cruz, Bernarda Henríquez, Margaret Baldwin, Benn Sartorius, Adam Kucharski, Eric James Nilles, Colleen L. Lau

**Affiliations:** 1School of Public Health, Faculty of Medicine, The University of Queensland, Brisbane, QLD 4006, Australia; a.cadavidrestrepo@uq.edu.au (A.C.R.); h.mayfield@uq.edu.au (H.J.M.); b.sartorius@uq.edu.au (B.S.); colleen.lau@uq.edu.au (C.L.L.); 2Ministry of Health and Social Assistance, Santo Domingo 10514, Dominican Republicfarahninoska@gmail.com (F.P.);; 3Brigham and Women’s Hospital, Boston, MA 02115, USAmgam81@gmail.com (G.A.); enilles@bwh.harvard.edu (E.J.N.); 4Infectious Diseases and Epidemics Program, Harvard Humanitarian Initiative, Cambridge, MA 02138, USA; 5Faculty of Health Sciences, Pedro Henriquez Urena National University, Santo Domingo 10514, Dominican Republic; wduke@unphu.edu.do; 6Harvard Medical School, Boston, MA 02115, USA; 7Centers for Disease Control and Prevention, Central America Regional Office, Guatemala City 01015, Guatemalafdx8@cdc.gov (B.L.); 8Department of Infectious Disease Epidemiology and Dynamics, Faculty of Epidemiology and Population Health, London School of Hygiene & Tropical Medicine, London WC1E 7HT, UK; adam.kucharski@lshtm.ac.uk

**Keywords:** SARS-CoV-2, seroprevalence, vaccine, spatial distribution

## Abstract

Incidence of COVID-19 has been associated with sociodemographic factors. We investigated variations in SARS-CoV-2 seroprevalence at sub-national levels in the Dominican Republic and assessed potential factors influencing variation in regional-level seroprevalence. Data were collected in a three-stage cross-sectional national serosurvey from June to October 2021. Seroprevalence of antibodies against the SARS-CoV-2 spike protein (anti-S) was estimated and adjusted for selection probability, age, and sex. Multilevel logistic regression was used to estimate the effect of covariates on seropositivity for anti-S and correlates of 80% protection (PT_80_) against symptomatic infection for the ancestral and Delta strains. A total of 6683 participants from 134 clusters in all 10 regions were enrolled. Anti-S, PT80 for the ancestral and Delta strains odds ratio varied across regions, Enriquillo presented significant higher odds for all outcomes compared with Yuma. Compared to being unvaccinated, receiving ≥2 doses of COVID-19 vaccine was associated with a significantly higher odds of anti-S positivity (OR 85.94, [10.95–674.33]) and PT_80_ for the ancestral (OR 4.78, [2.15–10.62]) and Delta strains (OR 3.08, [1.57–9.65]) nationally and also for each region. Our results can help inform regional-level public health response, such as strategies to increase vaccination coverage in areas with low population immunity against currently circulating strains.

## 1. Introduction

Latin America and the Caribbean Islands have been heavily affected by COVID-19 (SARS-CoV-2) [1]. The region was at times identified as the epicentre of the pandemic, and in mid-2022, accounted for 27% of COVID-19 deaths worldwide [2]. Among the Caribbean Islands, the Dominican Republic (DR) had reported 647,205 cases (24 October 2022), one of the highest cumulative numbers of COVID-19 cases at the time, second only to Cuba [3]. In the DR, the first cases of COVID-19 were reported in March 2020. The Delta strain was first detected in December 2020 and had become the dominant strain by August 2021. In December 2021, the Omicron strain was first detected and was dominant until February 2022 [4].

Modelled estimates suggested that by the end of 2020, 41.9% (95% CI 38.0–46.1%) of the population in the Americas had been infected with SARS-CoV-2 [5]. We previously reported findings from a national serological survey conducted in the DR between June and October 2021 and estimated that 77.5% of the population had been infected with SARS-CoV-2, and 89.5% of the population over 5 years of age had been immunologically exposed via infection, vaccination, or both [6].

Serological surveys have generated key data on the epidemiology and transmission of SARS-CoV-2 during the pandemic [6,7]. These studies have contributed to overcoming some of the challenges faced in COVID-19 surveillance, such as providing estimates for the proportion of mild or asymptomatic cases [8], lack of consistent diagnostic tests, and inaccurate case reporting due to overloaded health systems [9]. However, these studies are limited by uncertainties about the long-term immune protection against SARS-CoV-2 [10], the inability of simple serological tests to determine whether anti-S resulted from vaccination or infection [11], and the continuing emergence of new strains [12].

Infectious disease transmission is frequently spatially heterogeneous, even within the same country [13]. Geographical variations in the reported incidence of COVID-19 have been found to be influenced by sociodemographic factors such as urban settings [14], living in socioeconomically disadvantaged neighbourhoods [15], ethnic minority status [16], access to SARS-CoV-2 testing and hospitalization [15], shared accommodation [16], and number of household members [17]. Non-pharmaceutical interventions (NPIs) and vaccination campaigns are key strategies to control transmission [18]. However, vaccine uptake and the impact of non-pharmaceutical measures may vary as a result of cultural, political, and economic differences between affected areas, which make disease distribution even more varied. Therefore, a better understanding of the drivers of infection at different spatial scales could contribute to the identification of high-risk areas and provide information for more targeted implementation of control measures.

From March 2020, the DR implemented NPIs such as sanitary and epidemiological controls, restriction of mobility and social activities, closure of borders and ports, suspension of classes, and limitation of productive activities and public transport [19]. A national COVID-19 vaccination campaign was launched in February 2021 [19], and by August 2021, 52.3% of the population had received at least one dose of vaccine, 36.2% had received two doses, and 5.3% had received three doses [6]. In the DR, the inactivated vaccine CoronaVac^®^ (Sinovac Biotech, Beijing, China) was the primary vaccine adopted, representing almost 90% of all doses administrated through August 2021 [6].

Analysis of serological data at different spatial scales can help us characterise and understand the transmission dynamics and burden of SARS-CoV-2 [20]. Precise information on local seroprevalence and associated determinants can identify areas of low population immunity and highlight locations that are more vulnerable to future outbreaks and where public health interventions should be prioritized [20] However, evidence on spatial variation of SARS-CoV-2 seroprevalence at subnational levels has so far been limited.

Previously, we presented national-level seroprevalence and protective immunity against SARS-CoV-2 in the DR and identified risk factors associated with anti-S prevalence and correlates of protection at the country level [6]. Here we aim to investigate if anti-S seroprevalence is heterogeneously distributed among the regions, explore regional differences in anti-S prevalence and protective immunity, and identify regional variations in factors associated with anti-S prevalence and correlates of protection.

## 2. Materials and Methods

### 2.1. Setting

The DR is a Latin American country located in the Caribbean that shares the island of Hispaniola with Haiti. With ~10.5 million residents [21], it is the second most populous country in the Caribbean. The country is divided into 31 provinces plus the Santo Domingo National District, 155 municipalities, 386 district municipalities, 1565 sections, and 12,565 barrios/parajes. Provinces are aggregated into 10 administrative regions (Figure 1): Cibao Norte, Cibao Sur, Cibao Nordeste, Cibao Noroeste, Valdesia, Enriquillo, El Valle, Yuma, Higuamo, and Metropolitana [22]. The country’s division into regions reflects characteristics beyond the administrative reasons, as regions share environmental, sociodemographic, and historical characteristics. National data regarding population, economics, education, and health events are generally reported by region [22].

In the DR, 80% of the population live in urban and semi-urban areas, yet only about 20% of the total barrios/parajes are classified as urban setting. The metropolitan area of the capital, the Santo Domingo National District, is home to a population of 3.3 million people and is situated in the region of Metropolitana. Over the last two decades, the DR has experienced consistent economic growth, with an overall reduction in poverty [23]. However, social inequities remain, with higher levels of poverty in urban slums and rural areas, particularly in provinces close to the Haitian border [23].

### 2.2. Data Source, Study Design, and Study Procedures

National and provincial demographic data and cluster population and classification (urban versus rural) were obtained from the Dominican Republic National Statistics Office and the United National Statistics Division [21,24].

Individual level data used in this study were obtained from a three-stage cross-sectional national serosurvey conducted between 30 June and 12 October 2021 in 134 clusters (barrio/paraje) across the DR. A full description of sampling methods and national-level results have been described previously [6]. In summary, the 31 provinces plus the Santo Domingo National District were divided into 5 areas for logistical reasons, and within each area, a predefined number of urban and rural clusters were selected using a spatially representative sampling method [6]. In the selected urban clusters, a grid method was implemented for household selection [24]. In rural clusters, households were selected using a spatially representative sampling method that maximised spatial dispersion of sampling locations to ensure that we did not oversample both more populous areas and more sparsely populated areas [25]. A total of 23 households per cluster were selected in 132 clusters. Because the two remaining clusters were linked to a study of clinical surveillance of acute febrile illnesses [26], they were oversampled with 60 households per cluster. All household members aged ≥ 5 years were invited to participate.

A trained field team interviewed participants and collected questionnaire data using the Kobo Toolbox software (version 2.021.21, accessed from October 2020 to October 2021 - www.kobotoolbox.org). Global positioning system (GPS) coordinates of each household were captured using smartphones. Venous blood was collected from participants, processed as sera, and frozen at −80 °C. The samples were tested for anti-S antibodies using Roche Elecsys SARS-CoV-2 electrochemiluminescence immunoassays (Roche Diagnostics, Indianapolis, IN, USA). Previous studies have indicated robust assay performance, specificity of 99.8% (CI 99.3–100), and sensitivity of 98.2% (CI 96.5–99.2) [27,28]. Pseudoviral neutralization titres (PVNT) were used to estimate correlates of protective immunity against symptomatic infection for the ancestral and Delta strains. A PVNT of approximately 20% and 80% of mean convalescent titre is estimated to provide 50% and 80% protection against symptomatic infection, respectively, as previously described [6]. Individual-level PVNT was estimated by random forest binary classification.

Written consent was obtained from all participants. For children < 18 years old, except emancipated minors, consent was obtained from the legal guardian. Written assent was provided by adolescents 14–17 years old, and verbal assent was provided by children 7–13 years old. The study protocol was approved by the National Council of Bioethics in Health, Santo Domingo (013-2019), the Institutional Review Board of Pedro Henríquez Ureña National University in Santo Domingo, and the Mass General Brigham Human Research Committee, Boston, USA (2019P000094). The study was registered at the Human Research Ethics Committee of the University of Queensland (2022/HE001475). Study procedures and reporting adhered to the ROSES-S statement on reporting of sero-epidemiologic studies for SARS-CoV-2 [29].

### 2.3. Statistical Analysis

The statistical programming language R (R version 4.1.3, 2022-03-10) [30] was used to estimate prevalence accounting for sampling design (survey package) and for visualization of the data (ggplot2 and ComplexHeatmap). The national and regional multilevel survey-weighted mixed effects logistic regression models (accounting for the sampling design and weights) were run in Stata [31], using the generalized linear latent and mixed models (GLLAMM) package [32], to estimate the association of key determinants of antibody status. Prevalence distribution and kernel density maps were generated using Esri^®^ ArcGIS software v 10.8 (Esri^®^ ArcMap 10.8.0.12790. Redlands, CA, USA) [33]. 

#### 2.3.1. Anti-S Prevalence

Seroprevalence of anti-S was estimated at regional and cluster levels (134 clusters). Due to sample size, Cibao Sur was combined with Cibao Nordeste, and El Valle with Cibao Noroeste, resulting in 8 final regions used for analysis). At the regional level, anti-S seroprevalence was adjusted for selection probability and sampling design and post-stratified by age and sex. At the cluster level, seroprevalence was post-stratified by sex and age. The weights were corrected for finite population. For a full description, see Appendix A.

#### 2.3.2. Correlates of Protective Immunity against Symptomatic Infection

The results of the individual-level PVNTs were used to estimate the population’s immunological 80% protection against symptomatic infection (PT_80_) at the regional and cluster levels. At the regional level, PT_80_ was adjusted for selection probability and sampling design and post-stratified by age and sex. At the cluster level, PT_80_ was post-stratified by sex and age. The weights were corrected for finite population, as described above.

#### 2.3.3. Logistic Regression

Logistic regression analyses were conducted to assess the association between covariates and three outcomes of interest at national and regional scales: anti-S prevalence, PT_80_ for the ancestral strain, and PT_80_ for the Delta strain of SARS-CoV-2. Associations between outcomes and covariates were considered statistically significant if the 95% confidence interval (95% CI) of the estimated odds ratio (OR) excluded one. The covariates considered for the analyses included gender (male, female, other), age (categorized into 5–17, 18–54, ≥55 years old), educational level (categorized into technical/university level and none/primary/secondary level; for participants without educational level reported, the highest educational level in the household was assumed; in cases where no members of the household reported their educational level, educational level was assumed to be non-tertiary for the binary classification), socioeconomic score (categorized into zero or one, based on the access to city water, home toilet, household insect screens, air conditioning, and vehicle ownership; participants scored one point for any positive response and zero in the absence of a positive response), area of residence (rural or urban), number of household members (categorized into 1–2, 3–4, ≥5 people per household), work environment (outdoor, indoor, mix of indoor/outdoor combined, not active worker), smoking status (non-smoker, current smoker), comorbidities/risk factors (none and one or more of the following chronic diseases were considered risk factors: blood pressure, coronary heart disease, diabetes, cancer, kidney disease, previous stroke, chronic obstructive pulmonary disease, immunodeficiency), and doses of COVID-19 vaccines received (0, 1, ≥2). The variables included in the model were tested for multicollinearity using the variance inflation factor (VIF) with a cut-off value of <3.

We previously reported results of standard multivariable logistic regression models for anti-S, and for the PT_80_ for the ancestral and Delta strains, incorporating gender, age, area of residence, number of household members, work environment, smoking status, and number of COVID-19 vaccine doses as covariates [5]. In this study, we built on the previous analyses by examining regional differences in potential drivers of prevalence distribution and additional sociodemographic factors listed earlier. Two different models were built as follows:A national multivariable model that includes the ten regions (administrative divisions) as a covariate, as well as the sociodemographic covariates indicated above. The weights, calculated based on the survey design, were incorporated into the regression. This model was built to assess differences in ORs between regions.Multilevel survey-weighted mixed effects logistic regression model fitted at national and regional levels:2.1The national level included region, cluster, and household as random effects, gender, age, area of residence, number of household members, work environment, smoking status, educational level, socioeconomic score, comorbidities/risk factors, and number of vaccine doses as fixed effects. To account for the sampling design, the weights of the selection probability were calculated in three stages. First, the probability of a cluster being selected (pc) was calculated based on the total number of clusters in each category and the weight (wc) was the inverse of the probability of selection for each category (wc = 1/pc). Second, the probability of a household being selected (ph) was calculated based on the total number of households in each cluster and the weight (wh) was the inverse of the probability of household selection (wh = 1/ph). Third, the weights (wf) from the first two steps were multiplied (wf = wc × wp) and corrected for a finite population. The full description of the weight calculation can be found in the Appendix A.2.2The regional level included cluster and household as random effects, gender, age, area of residence, number of household members, work environment, smoking status, educational level, socioeconomic score, comorbidities/risk factors, and number of vaccine doses as fixed effects, and incorporated level-specific sampling weights to account for sampling design. This model was built to identify variations in strength and significance of association across regions. Due to sample size, Cibao Sur was combined with Cibao Nordeste, and El Valle with Cibao Noroeste.

#### 2.3.4. Kernel Density Maps

Cluster-level prevalence of anti-S and PT_80_ for the ancestral and Delta strains were used to calculate a magnitude-per-unit area, and via a non-parametric estimation technique, a smoothed surface was fit to the original data to generate kernel density maps. We used the outcome with the greatest variation in prevalence to determine the categories used in the intervals of distribution. Those values were adopted as a reference when building maps of the other outcomes, to allow comparison between the distribution of the different outcomes. Kernel density maps were used to estimate the distribution of outcomes in unsampled areas and visualise spatial variation in outcome distribution.

## 3. Results

### 3.1. Participants and Demographics

This study included data from 6683 participants (84.4% of the 7916 eligible individuals present at the time of household visit) from 3832 households in 134 clusters across all 10 regions of the DR. A total of 4171 (62.4%) of the participants were female, and 4959 (74.2%) were aged 15–64 years (mean 41.4 years, standard deviation [SD] 20.5 years). Within each region, the gender and age distributions reflect national census data (Table 1). Participants living in urban areas were 53.8% of the total included in this study, varying from 40.3% to 67.5% across regions. The number of households with 3–6 members varied from 65.5% to 71.8% among regions.

Overall, 7.0% of all participants were current smokers (10.1% among males and 4.9% among females). The prevalence of current smokers varied between regions, from 5.7% in Higuamo to 11.8% in Cibao Noroeste/El Valle.

### 3.2. Anti-S Seroprevalence and PT_80_ for the Ancestral and Delta Strains

The adjusted anti-S prevalence at the national level is shown in Appendix A. At the national level, the overall adjusted anti-S prevalence was 85.4% (95% CI 81.9–88.0%), with the lowest prevalence (69.9%, 95% CI 65.9–73.7%) in the age-group 5–17 years, and the highest prevalence in participants who had received two or more doses (98.9%, 95% CI 97.9–99.5%) of COVID-19 vaccine.

Figure 2 provides a geographic summary of the adjusted anti-S prevalence, PT_80_ for the ancestral and Delta strains, and COVID-19 vaccine coverage (two or more doses). The top row presents results at the regional level, and the second row at the cluster level. The third row shows kernel density maps of each outcome, and the last row shows the density distribution among clusters.

The adjusted prevalence of anti-S among individuals aged ≥5 years varied between regions. The lowest prevalence was observed in Yuma, at 78.7% (95% CI 75.0–82.2%), in the east, and the highest in Enriquillo, 90.4% (95% CI 86.1–93.8%), in the southwest, bordering Haiti (Appendix A). At the cluster level, anti-S prevalence varied from 34.3% (95% CI 32.6–36.0%) to 98.9% (95% CI 98.5–99.3%; see Appendix A). In 46 clusters (34.3%), anti-S prevalence estimates were above 90%. Five clusters presented a crude prevalence of 100%, and it was not possible to estimate adjusted prevalence and 95% CI. 

The adjusted proportion of the population aged ≥ 5 years estimated to have at least 80% protection against symptomatic infection (PT_80_) for the ancestral strain varied among regions; the lowest estimate was observed in Yuma at 57.7% (95% CI 54.6–60.9%), and the highest in Enriquillo, at 72.0% (95% CI 70.7–73.3%) (Appendix A). Among clusters, PT_80_ for the ancestral strain varied from 10.5% (95% CI 9.2–11.7%) to 95.5% (95% CI 94.7–96.23%). 

The adjusted proportion of the population aged ≥ 5 years estimated to have PT_80_ against the Delta strain also varied between regions; the lowest estimate was observed in Yuma at 30.0% (95% CI 27.8–32.2%), and the highest in Enriquillo at 43.0% (95% CI 37.1–49.0%—Appendix A). Among clusters, PT_80_ for the Delta strain varied from 1.0% (95% CI 0.7–1.3%) to 66.7% (95% CI 65.5–-67.8%) (Appendix A). 

The proportion of participants that had received two or more doses of the COVID-19 vaccine varied among regions and clusters. Yuma had the lowest vaccination coverage at 22.1%, (95% CI 16.7–28.1%), and Higuamo the highest (41.5%, 95% CI 31.0–52.6%) (Appendix A). 

### 3.3. Logistic Regression

#### 3.3.1. Multivariable Model for Anti-S, and PT_80_ for the Ancestral and Delta Strains 

No multicollinearity was detected between variables. Results from the multivariable logistic regression models for the three outcomes of interest (anti-S, PT_80_ for the ancestral strain, PT_80_ for the Delta strain) were used to determine if there were significant differences in prevalence among the regions. The findings are summarised in Figure 3. Yuma was used as the reference region for the model, as it had the lowest anti-S prevalence. The results show that the odds of having an anti-S positive result or PT_80_ for the ancestral and Delta strains varied between regions; however, statistically significant differences were not observed within most regions. The exceptions were Anti-S and PT_80_ for the ancestral strain in the Metropolitana region, PT_80_ for the ancestral and Delta strains in El Valle, and Anti-S, PT_80_ for the ancestral and Delta strain in Enriquillo (OR 1.63; 95% CI 1.0–2.64). 

Other significant results as described in the Appendix A.

#### 3.3.2. Multilevel Logistic Regression Models

##### National Models

Results from the national multilevel model for anti-S positivity are shown in Figure 4. Compared to those aged 18–54 years, those aged 5–17 years had an OR of 0.17 (95% CI 0.07–0.40). The OR of people working in indoor/mix of indoor and outdoor environment was 9.49 (95% CI 1.33–67.85) compared to people working in outdoor environments. Compared to those responding none/primary/secondary in terms of educational level, people with tertiary/technical education had an OR of 1.78 (95% CI 1.39–2.29). Compared to unvaccinated people, those who received one dose or two or more doses had ORs of 4.80 (95% CI 2.35–9.83) and 85.94 (95% CI 10.95–674.33), respectively. 

The results of the national multilevel models for PT_80_, for both the ancestral and Delta strains, also identified vaccination as the strongest predictor of a positive result (Appendix A). For the ancestral strain, those respondents having one or two or more doses had ORs of 2.61 (95% CI 1.72–3.94), 4.78 (95% CI 2.15–10.62), respectively. For the Delta strain, those respondents having two or more doses had an OR of 3.08 (95% CI 1.57–9.65). 

Besides vaccination, results from the national models identified other factors associated with higher odds for PT_80_ for the ancestral and Delta strains. For the ancestral strain, higher odds were seen in those working in indoor/mix of indoor/outdoor environment OR 4.34 (95% CI 2.43–7.76) compared to outdoor work environment. For the Delta strain, those aged ≥ 55 years, the OR was 1.45 (95% CI 1.11–1.89) compared to those aged 18–54 years and indoor/mix of indoor/outdoor work environment had an OR of 2.41 (95% CI 1.16–4.97) compared to outdoor work environment. 

##### Regional Models

Despite the relatively small sample size per region, those who received two or more doses of the COVID-19 vaccine had significantly higher ORs when compared to the unvaccinated for anti-S and PT_80_ for the ancestral and Delta strains in all regions (Figure 5A–C). Overall, the results of the multilevel logistic regression models for all outcomes in each region were not statistically significant for most covariates (Appendix A). However, it showed trends regarding the association of covariates in each region (i.e., higher odds associated with vaccination status and area of residence and lower odds associated with younger age groups). Anti-S had significantly lower ORs for participants under 18 years compared with those aged 18–54 years in most regions, except Enriquillo and Valdesia (Appendix A). Other significant results for all outcomes can be found in the Appendix A.

## 4. Discussion

To our knowledge, this is the first study to assess the spatial variation in anti-S seroprevalence at the regional and cluster levels and to evaluate factors associated with protection against symptomatic SARS-CoV-2 infection at the sub-national level in the DR. The findings suggest that the prevalence of anti-S and PT_80_ for the ancestral and Delta strains were spatially heterogeneous at regional and cluster levels and highlight areas of low population immunity that may be more vulnerable to potential future outbreaks. Two or more doses of the COVID-19 vaccine had the strongest association with higher odds of anti-S positivity and correlates of protection at the national and regional levels. We also quantified the relative importance of sociodemographic covariates on anti-S positivity and correlates of protection. This information may help guide policymakers in implementing vaccination campaigns and targeting regions/communities with lower current vaccine coverage and particular sociodemographic sub-populations with lower immunity. These models can be used for various scenario analysis of disease transmission and immunity at subnational scale to identify and forecast population sub-groups at greatest risk for new COVID infection and related morbidity/mortality, waning immunity etc.

The regional differences observed in this study contribute to a better understanding of the distribution of the COVID-19 burden in the DR. A previous study, conducted in the DR between April and June 2020, identified heterogeneous distribution of immunoglobulin IgM and IgG across ten emerging hotspots of transmission and indicated that pathogen circulation preceded community-level interventions [34]. Two hotspots located in the touristic region of Yuma presented low seroprevalence compared to hotspots located in Cibao Norte and Nordeste. This difference remained present in our study, in which Yuma had the lowest anti-S seroprevalence, PT_80_ for the ancestral strain, and vaccine coverage. Our findings suggest that this region could be prone to a higher risk of an outbreak, particularly if the regional differences identified in this study persist over time. Enriquillo had the highest prevalence of anti-S and correlates of protection; this region is located on the border with Haiti and has the highest percentage of participants living in urban areas and of households with five or more members. The Enriquillo region also had the highest odds of anti-S positivity, PT_80_ for the Delta strain, and the second highest odds of PT_80_ for the ancestral strain, confirming regional differences in the COVID-19 burden across regions. 

Although these regional differences had an impact on SARS-CoV-2 distribution, our findings suggest a greater importance of factors that might occur on a smaller spatial scale. A higher risk of transmission in enclosed environments or between household members, when compared with the risk of transmission between community members, has been identified [35,36]. In agreement with these findings, all our national models consistently identified two factors associated with higher odds of positivity: indoor and mixed work environments and COVID-19 vaccination. Conditions that facilitate the spread of viral particles via aerosol and droplets, such as poor ventilation, difficulty in maintaining distance from a sick individual, and longer exposure time, might explain the relevance of indoor work environments [37,38,39,40,41]. These findings support our previous report at the national level [6]. 

Vaccination has so far been the most efficient public health intervention for preventing symptomatic infections and severe COVID-19 [42,43,44,45]. However, the emergence of new strains has raised concern about COVID-19 vaccine efficacy and lasting immunity generated via prior infection, with demonstrably lower protection against symptomatic infection against the highly immune-evasive Omicron strains [46]. The principal COVID-19 vaccines administered in the DR were the inactivated viral CoronaVac (Sinovac, Biotech, Beijing, China), the adenovirus vector ChAdOx1-S (Oxford/AstraZeneca, Cambridge, United Kingdom), and the mRNA BNT162b2 (Pfizer/BioNTech, New York, USA). These vaccines had their effectiveness against the Delta strain tested both for symptomatic and severe infection [47,48,49], with better performance on individuals that received the third dose [50,51]. Our results from the PT_80_ for the Delta strain models support the importance of the third dose to develop protective immunity. 

Results from the separate multilevel models for each region were not uniform, indicating that, except for vaccination, national results might not be generalized to sub-national levels at the time of this study. Indoor work environment was a key factor in the national models but was not associated with higher odds in the regional models. This effect might be related to a more even rate of indoor and outdoor work environments in each region [23]. 

The strengths of this study include the robust sampling design that was able to provide a well-distributed and nationally representative sample, as it accounted for characteristics of population distribution, such as the differences between rural and urban areas and a higher concentration on two large urban areas. The different models were able to quantify the relative importance of drivers and generate data that might be useful for future modelling of COVID-19 transmission. Part of the limitations of this study is related to the risk of misinterpretation regarding the anti-S positive result. The detection of anti-S can be indistinguishably related to previous infection or vaccination [52], which are not mutually exclusive and may be associated with different drivers [53]. Moreover, seroprevalence should not be interpreted as immunity, which would be an oversimplification of the complex immune response [54]. Depending on the type of exposure to SARS-CoV-2 strains, antibodies can wane over time [54], and factors such as age [55], comorbidities [56], and smoking [57] can also influence antibody production. In our study, smoking history was limited to “current smoker”, “prior smoker”, and “never smoked” and current smokers were underrepresented compared to previous studies [58], which may have prevented the accurate identification of associations between tobacco use and seroprevalence of SARS-CoV-2. Cellular immune response could not be assessed through this seroprevalence survey. The Omicron strain was detected in December 2021, and, in this study, samples were not tested for neutralizing activity against Omicron. Studies that investigated the effectiveness of the three main vaccines used in the DR against the Omicron strain indicated that regardless of the mild protection against symptomatic disease, the protection against severe disease remained high [41,42,43,44,59,60,61]. Lastly, the relatively small sample size per region may have limited the identification of significant variation of covariates importance across regions. 

The results presented here can inform more targeted regional-level public health response by identifying areas with low population immunity, which might be more prone to severe outbreaks and will also benefit the most from increased vaccination coverage and other public health measures. It is likely that SARS-CoV-2 will continue to circulate endemically for years and that not all individuals present the same susceptibility. Although our study was based on cross-sectional data, the regional differences in risk factors and drivers of protective immunity will likely persist over time (e.g., regions with relatively low vaccination coverage will likely remain low). Therefore, the areas identified in our study that will benefit most from targeted public health measures, such as vaccination, will likely continue to be relevant.

## Figures and Tables

**Figure 1 tropicalmed-08-00493-f001:**
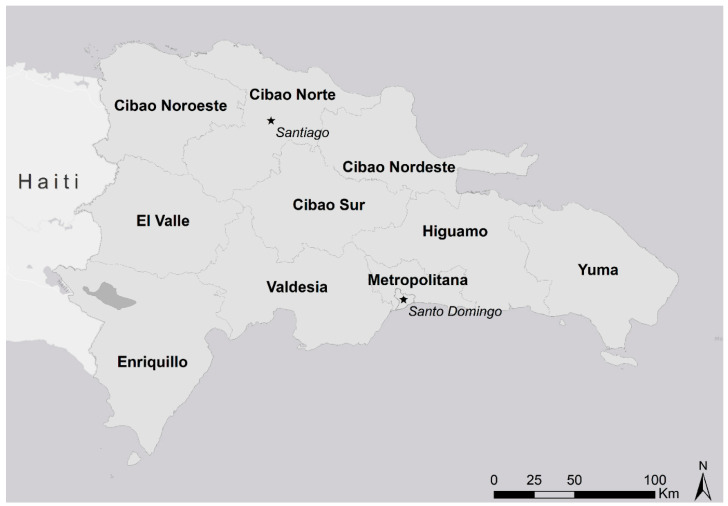
Administrative regions of the Dominican Republic. The stars are showing Santiago Province (North) and Santo Domingo Capital district (Southeast). Base map downloaded from: Esri, HERE, Garmin, ©OpenStreetMap contributors, and the GIS user community (available on: https://basemaps.arcgis.com/arcgis/rest/services/World_Basemap_v2/VectorTileServer, accessed on 28 October 2023).

**Figure 2 tropicalmed-08-00493-f002:**
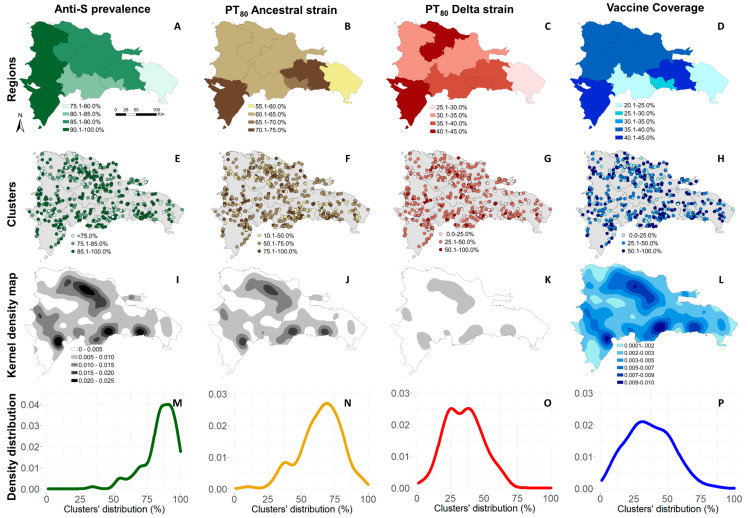
Adjusted anti-S prevalence, correlates of protection for the ancestral and Delta strains, and vaccine coverage in the Dominican Republic, June–October 2021. (**A**–**D**) Anti-S prevalence, PT_80_ for the ancestral strain, PT_80_ for delta strain, COVID-19 vaccine coverage (two or more doses) by region. (**E**–**H**) Anti-S prevalence, PT_80_ for the ancestral strain, PT_80_ for the Delta strain and COVID-19 vaccine coverage (two or more doses) by cluster. (**I**–**L**) Kernel density maps of anti-S prevalence, PT_80_ for the ancestral strain, PT_80_ for the Delta strain, and COVID-19 vaccine coverage (two or more doses). (**M**–**P**) Anti-S prevalence, PT_80_ for the ancestral strain, PT_80_ for delta strain, COVID-19 vaccine coverage (two or more doses) density distribution among clusters.

**Figure 3 tropicalmed-08-00493-f003:**
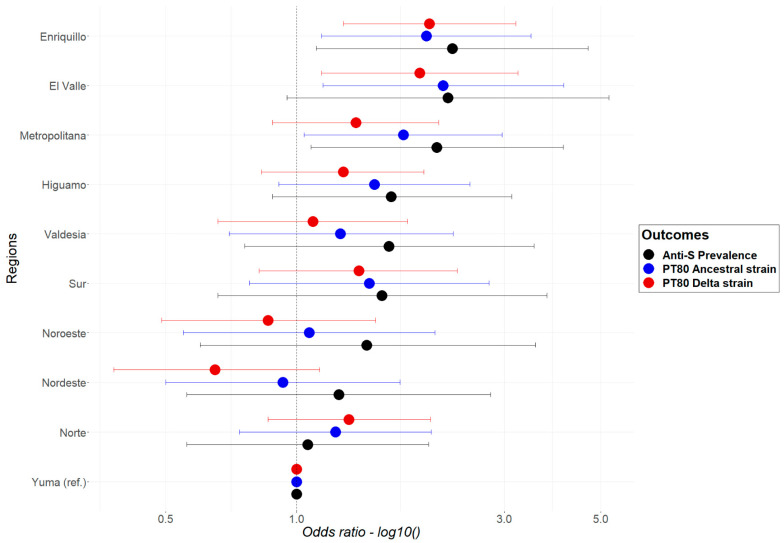
Odds ratio and 95% CI from the multivariable logistic regression, for anti-S (black), PT_80_ for the ancestral (blue), and for the Delta (red) strains of the SARS-CoV-2 by region, in the Dominican Republic, June–October 2021.

**Figure 4 tropicalmed-08-00493-f004:**
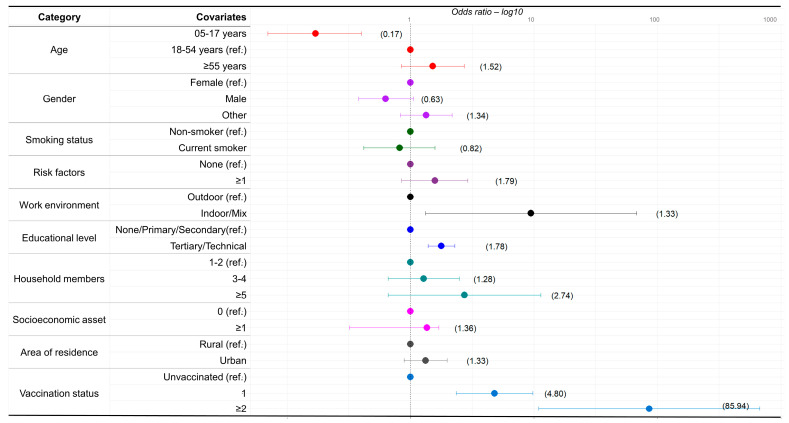
National multilevel logistic regression model. Covariates associated with anti-S positivity in the Dominican Republic, June–October 2021.

**Figure 5 tropicalmed-08-00493-f005:**
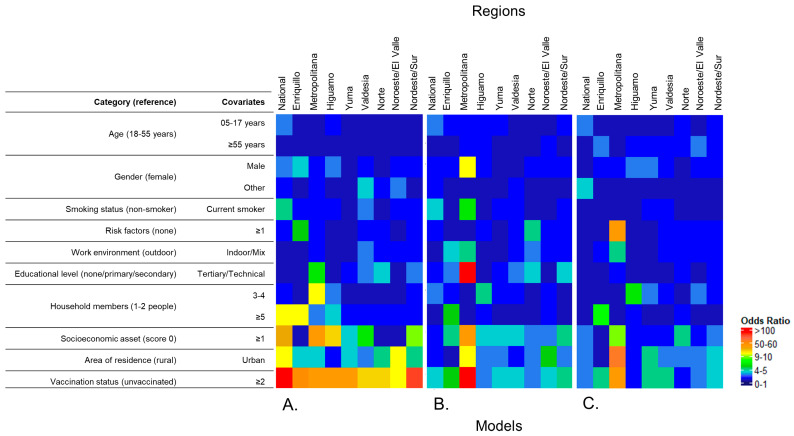
Regional heatmaps for odds of anti-S positivity, PT_80_ for the ancestral and Delta strains of the SARS-CoV-2, from regional multilevel logistic regression, Dominican Republic, June–October 2021. To construct the heatmap, odds ratios were classified as follows: 0.1–1.0, 1.1–2.0, 2.1–3.0, …, 9.1–10.0, 10.1–20.0, 20.1–30.0, 30.1–40.0, …, 90–100.0, ≥100.0. (**A**) Odds ratios from the Anti-S model. (**B**) Odds ratios from PT_80_ for the ancestral strain model. (**C**) Odds ratios from PT_80_ for the Delta strain model.

**Table 1 tropicalmed-08-00493-t001:** Demographic characteristics and COVID-19 vaccination status of study participants by region and nationally, Dominican Republic, June–October 2021.

	Yuma 330 (%)	Nordeste/Sur 533 (%)	Valdesia 489 (%)	Noroeste/El Valle 602 (%)	Norte 1306 (%)	Higuamo 1717 (%)	Metropolitana 999 (%)	Enriquillo 707 (%)	National 6683 (%)
**Gender**									
Female	128 (38.8)	185 (34.7)	188 (38.4)	234 (38.9)	489 (37.4)	602 (35.1)	391 (39.1)	277 (39.2)	2512 (37.6)
Male	198 (60)	347 (65.1)	295 (60.3)	367 (61)	811 (62.1)	1099 (64.0)	598 (59.9)	429 (60.7)	4171 (62.4)
Other	4 (1.2)	1 (0.2)	6 (1.2)	1 (0.2)	6 (0.5)	16 (0.9)	10 (1.0)	1 (0.1)	45 (0.7)
**Age**									
05–17 y	47 (14.2)	25 (4.7)	62 (12.7)	78 (13.0)	120 (9.2)	274 (16.0)	164 (16.4)	142 (20.1)	912 (13.6)
18–54 y	183 (55.5)	295 (55.3)	286 (58.5)	328 (54.5)	745 (57.0)	1007 (58.6)	553 (55.4)	398 (56.3)	3975 (59.5)
>55 y	100 (30.3)	213 (40)	141 (28.8)	196 (32.6)	441 (33.8)	436 (25.4)	282 (28.2)	167 (23.6)	1976 (29.6)
**Educational level**									
None/Primary/Secondary	288 (87.3)	390 (73.2)	343 (70.1)	438 (72.8)	1059 (81.1)	1406 (81.9)	733 (73.4)	539 (76.2)	5196 (77.7)
Tertiary/Technical	42 (12.7)	143 (26.8)	146 (29.9)	164 (27.2)	247 (18.9)	311 (18.1)	266 (26.6)	168 (23.8)	1487 (22.3)
**Socioeconomic score**									
0 pts	33 (10.0)	6 (1.1)	6 (1.2)	18 (3.0)	26 (2.0)	81 (4.7)	37 (3.7)	78 (11.0)	285 (4.3)
1–5 pts	294 (89.1)	521 (97.7)	474 (96.9)	581 (96.5)	1263 (96.7)	1624 (94.6)	952 (95.3)	628 (88.8)	6337 (94.8)
**Area of residence**									
Rural	169 (51.2)	245 (46.0)	259 (53.0)	288 (47.8)	732 (56.0)	729 (42.5)	434 (43.4)	230 (32.5)	3086 (46.2)
Urban	161 (48.8)	288 (54.0)	230 (47.0)	560 (93.0)	574 (44.0)	988 (57.5)	565 (56.6)	477 (67.5)	3597 (53.8)
**Household members**									
1–2p	64 (19.4)	163 (30.6)	57 (11.7)	106 (17.6)	335 (25.7)	373 (21.7)	210 (21.0)	119 (16.8)	1427 (21.4)
3–4p	133 (40.3)	232 (43.5)	206 (42.1)	250 (41.5)	579 (44.3)	716 (41.7)	407 (40.7)	253 (35.8)	2776 (41.5)
>5p	133 (40.3)	138 (25.9)	226 (46.2)	246 (40.9)	392 (30.0)	628 (36.6)	382 (38.2)	335 (47.4)	2480 (37.1)
**Work environment**									
Indoor/Mix	63 (19.1)	167 (31.3)	95 (19.4)	141 (23.4)	382 (29.2)	315 (18.3)	233 (23.3)	135 (19.1)	1531 (22.9)
Outdoor	38 (11.5)	38 (7.1)	53 (10.8)	52 (8.6)	68 (5.2)	123 (7.2)	90 (9.0)	65 (9.2)	527 (7.9)
**Smoking status**									
Current smoker	28 (8.5)	43 (8.1)	45 (9.2)	41 (6.8)	90 (6.9)	98 (5.7)	70 (7.0)	54 (7.6)	469 (7.0)
Non-smoker	302 (91.5)	490 (91.9)	444 (90.8)	561 (93.2)	1216 (93.1)	1619 (94.3)	929 (93.0)	653 (92.4)	6214 (93.0)
**Risk factors**									
None	189 (57.3)	253 (47.5)	278 (56.9)	331 (55.0)	663 (50.8)	1011 (58.9)	578 (57.9)	440 (62.2)	3743 (56.0)
>1	141 (42.7)	280 (52.5)	211 (43.1)	271 (45.0)	643 (49.2)	706 (41.1)	421 (42.1)	267 (37.8)	2940 (44.0)
**Vaccine doses**									
Unvaccinated	182 (55.2)	186 (34.9)	222 (45.4)	213 (35.4)	398 (30.5)	691 (40.2)	406 (40.6)	278 (39.3)	2576 (38.5)
1	53 (16.1)	123 (23.1)	81 (16.6)	90 (15.0)	176 (13.5)	213 (12.4)	143 (14.3)	73 (10.3)	952 (14.2)
>2	95 (28.8)	224 (42.0)	186 (38.0)	299 (49.7)	732 (56.0)	813 (47.4)	450 (45.0)	356 (50.4)	3155 (47.2)

## Data Availability

The data presented in this study are available on request from the corresponding author. The data are not publicly available due to preserving participants’ privacy.

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
