# Peer review of "Using Regional Sero-Epidemiology SARS-CoV-2 Anti-S Antibodies in the Dominican Republic to Inform Targeted Public Health Response"

_tropicalmed, 2023, doi:10.3390/tropicalmed8110493_

Round 1

Reviewer 1 Report

Comments and Suggestions for Authors

Congratulations on an interesting and well written study. I have the following comments to improve the report:

Introduction:

Line 53: by saying “these studies have been limited by…” makes it sound like your study now overcomes these issues. However, these are all still issues. Recommend to rephrase to ”these studies are limited by”

Methods:

Line 208: states “incorporated level-specific sampling weights to account for the sampling design”.  This is not detailed enough for the reader to understand if post-hoc clustering adjustment was conducted to account for selection bias in participant sampling.  As all 125 household members aged ≥5 years were invited to participate, this could elevate the seroprevalence estimate and broaden the confidence interval. Please describe more precisely the post hoc adjustments made and consider this adjustment if not already conducted.

Line 130: It is essential to include the diagnostic test kit used to test for anti-S antibodies. The sensitivity and specificity are important to know to determine accuracy of results. Also, post hoc test adjustment can also then be conducted to account for potential biases in estimates introduced by serological assay performance.  

Line 144: recommend to use the more specific SARS-COV-2 reporting guidelines for sero-epi studies: “ROSES-S: Statement from the World Health Organization on the reporting of seroepidemiologic studies for SARS-CoV-2”, https://onlinelibrary.wiley.com/doi/full/10.1111/irv.12870

Results:

In general, suggest to move some results to the supplement section to make this section more concise. 

Lines 288-290: this is discussion rather than your results so should be moved to that section.

Figure 5: recommend labelling the x-axes i.e. “region” for the top x-axis, “model” for bottom x-axis.

Discussion:

Line 373: this statement about opening boarders seems outdated now as tourism must have restarted. Recommend to update.

In general, the discussion should be strengthened to discuss results in comparison to other studies in the region. Recommend to compare results to those in meta-analyses e.g.  Bergeri et al. https://journals.plos.org/plosmedicine/article?id=10.1371/journal.pmed.1004107

Also check the Serotracker platform for any other studies in the region that found similar or disparate results and comment on this: https://serotracker.com/en/Explore

Reviewer 2 Report

Comments and Suggestions for Authors

Thank you for the opportunity to familiarize myself with your study.
I was unable to find a similar analysis in the Dominican Republic, which makes your research particularly interesting.

I have a few questions regarding the methods used for assessing antibody levels.

Were all individuals tested using the same commercial test?
If so, what was it, and what were its sensitivity and specificity?
My second question relates to the specifics of sample selection in various regions. Besides stratification by age and gender, were any other indicators used?
Additionally, a technical question: how was the question concerning cigarette smoking framed? Was the number of cigarettes smoked per day specified?

Once again, thank you for the opportunity to review this intriguing work. I confirm that the conclusions drawn from it will have an impact on the planning of future vaccination campaigns.

Reviewer 3 Report

Comments and Suggestions for Authors

The work is very well done, evaluating the regional seroepidemiology of SARS-CoV-2 antibodies in the Dominican Republic(DR) to inform a targeted public health response. As  reported by the authors   this is the first study to assess spatial variation in the seroprevalence of anti-S antibodies at a regional and cluster level and to evaluate factors associated with protection against symptomatic SARS-CoV-2 infection at a sub-national level in the DR.

Interestingly, the study also reports on the administration of COVID-19 vaccines in the DR. The vaccines administered were heterogeneous and the study presented is of great relevance.   These vaccines had their efficacy against the Delta strain tested in both symptomatic and severe infection, with better performance in individuals who received the third dose. The  results support the importance of the third dose in developing protective immunity in the population .

Although  the  study was based on cross-sectional data, regional differences in risk factors and drivers of protective immunity are likely to persist over time (for example, regions with relatively low vaccination coverage are likely to remain low). Therefore, the areas identified in the study that will benefit most from targeted public health measures such as vaccination are likely to remain relevant.

The study used appropriate sampling involving a population with different age groups. The techniques used are appropriate and have been carefully analysed . The study used appropriate sampling involving a population with different age groups. 
